# Pancreatic ductal adenocarcinoma: Prognostic indicators of advanced disease

**Deirdré Kruger** [1]☯*, **Nicola Lahoud**[1]☯, **Yandiswa Y. Yako** [1]¤, **John Devar**[1,2], **Martin Smith**[1,2]

1 Department of Surgery, School of Clinical Medicine, Faculty of Health Sciences, University of the Witwatersrand, Johannesburg, South Africa, 2 Hepato-Pancreatico-Biliary Unit, Department of General Surgery, Chris Hani Baragwanath Academic Hospital, Johannesburg, South Africa

☯ These authors contributed equally to this work.
¤ Current address: Department of Human Biology, Faculty of Health Sciences, Walter Sisulu University, Mthatha, South Africa
* Deirdre.Kruger@wits.ac.za

## Abstract

### Background/Objectives

Pancreatic ductal adenocarcinoma (PDAC) is an aggressive malignancy associated with high metastatic risk. Prognosis remains poor even after resection. Previously our group identified biomarkers that improved diagnostic accuracy in PDAC beyond the established diagnostic tumour marker, CA19-9. Risk factors, symptoms and circulating biomarkers associated with a PDAC diagnosis may differ from those that alter disease progression and metastasis. This study aimed at assessing the risk factors, presenting symptoms and potential prognostic biomarkers in PDAC and determine their relationship with PDAC stage and/or metastatic status.

### Methods

Seventy-two PDAC patients with imaging available for TNM staging at presentation were enrolled following informed consent. Demographic and clinical data were captured. Blood was collected and 38 cytokines/angiogenic factors measured. Nonparametric association tests, univariate and multivariate logistic regression were performed using STATA version 14.2. A p-value≤0.05 was considered significant and odds ratios reported for effect size.

### Results

Most risk factors and symptoms did not differ across the stages of cancer. Although male gender and smoking are risk factors for PDAC, the majority of study patients with metastatic PDAC were non-smoking females. In addition to CA19-9, the platelet count (p<0.01), IL-15 (p = 0.02) and GM-CSF (p<0.01) were significant, independent negative predictors of metastatic PDAC. Moreover, using specific cut-off values in a combined panel, the odds in a patient with all three biomarker levels below the cut-offs is 21 times more likely to have metastatic PDAC (p<0.0001).

**Data Availability Statement:** Data cannot be shared publicly because of restrictions from the Human Research Ethics Committee. Data are available from the University of the Witwatersrand Ethics Committee (contact via HREC-Medical.

ResearchOffice@wits.ac.za) for researchers who meet the criteria for access to confidential data.

**Funding:** MS. MRC CERCP15 Pancreas South African Medical Research Council www.samrc.ac.za MS conceived the study design, data interpretation, critically revised the manuscript and approved the final version.

**Competing interests:** The authors have declared that no competing interests exist.

## Conclusions

Platelet count, IL-15 and GM-CSF are potential prognostic indicators of metastatic disease in PDAC patients from our local South African population.

## Introduction

Pancreatic cancer is the ninth and tenth most common cancer globally in women and men, respectively, with the lowest survival rate for all stages combined when compared to other cancers [1]. As an aggressive malignancy which is associated with a high metastatic risk, only approximately 10–20% of patients are diagnosed when the tumour is resectable and overall the median survival is less than one year with mortality to incidence ratio unchanged over the past two decades. Moreover, median survival remains poor even after surgical resection [2–4].

Pancreatic cancer generally refers to pancreatic ductal adenocarcinoma (PDAC), accounting for approximately 85% of all pancreatic neoplasms [2]. Studies from the USA report considerably higher incidence rates of PDAC in Black patients than in any other racial group (1975–2014) and this is mirrored by mortality rates [5]. From the outdated South African National Cancer Registry (2014), a tissue based registry known for under-reporting, the number of pancreatic cancer patients diagnosed nationally was 167 males and 176 females (48.7% vs 51.3%, respectively), with the highest incidences in patients from Caucasian ethnicity amongst both genders (50.9% and 50.6%, respectively) [6]. The most frequent age range of diagnosis for both genders was 60–69 years.

### Risk factors for pancreatic cancer

The estimated lifetime risk of developing PDAC is relatively low. Nevertheless, unmodifiable risk factors for pancreatic cancer commonly include increasing age, male gender, Black race, family history of pancreatic cancer and inherited genetic syndromes. Modifiable risk factors mainly include obesity, diet, smoking and diabetes mellitus. Moreover, risk factors associated with a cancer diagnosis may differ substantially from those that modify cancer progression and survival. Furthermore, there is paucity in the literature on how specific risk factors impact on the progression and/or survival in different cancers. Where some data exist for breast, colon, prostate, and lung cancers [7], the impact of specific risk factors on PDAC progression have not been reported on.

### Symptoms for pancreatic cancer

Most patients with early PDAC are asymptomatic and initial symptoms are often vague, non-specific and intermittent. The latter is one of the main reasons why patients are initially falsely reassured that their symptoms cannot be of any importance [8]. Symptoms may differ depending on the location of the tumour in the head, body or tail of the pancreas. Overall, the most frequent and concerning symptoms at presentation include: [6, 9] abdominal pain, in up to 70% of patients in South Africa; asthenia, or lack of energy, reported to be prevalent in 30–86% of PDAC patients; obstructive jaundice, reported in 50–75% of PDAC; loss of a lot of weight reported in up to 85% of patients and more common with cancers in the head of the pancreas. Although some of these symptoms overlap for biliary tract cancers, unique features of PDAC in the two years prior to diagnosis were identified through a large multivariate primary healthcare database in the UK as back pain, lethargy and new onset adult diabetes [10].

Few studies have looked at symptoms and the duration of symptoms on PDAC disease progression, resectability and survival, and reportedly clinical presentation had no impact on

resectability or survival [11, 12]. No studies from Sub-Saharan Africa have investigated the association between symptoms and disease progression or prognosis in PDAC.

### Early diagnosis and progression of PDAC

Population-based screening for PDAC is not feasible due to the low incidence of this cancer in the general population and, moreover, due to the low overall prevalence the number of cases that would have to be screened to prevent one death is very high. Hence, PDAC is often undetected until it is an advanced stage. Early diagnosis depends on the effect of the mass and this depends on the location of the tumour within the pancreas. The early diagnosis of PDAC is further challenged by the absence of accurate diagnostic biomarkers. The individual power of the long established diagnostic tumour marker in PDAC, CA19-9, to accurately diagnose PDAC from healthy individuals or patients with benign disease is low and further influenced by the presence of obstructive jaundice and the host's inflammatory response [13–15]. In addition, where the role of platelets in tumour growth and metastasis has been reported in many cancer studies, the association between platelet counts and prognosis in PDAC patients has been inconclusive [16–21].

Recently, our group investigated the largest array of circulating angiogenic factors and inflammatory cytokines in any one study to date to determine whether combinations of these potential biomarkers could facilitate the earlier diagnosis of PDAC in the South African setting [22]. We reported novel combined biomarker panels that significantly improve diagnostic accuracy in PDAC. Specifically, the cytokines interferon-γ inducible protein (IP-10 or CXCL10), IL-8 and IL-15 are significant additions to biomarker panels in diagnosing PDAC in Black South African patients. The role of these cytokines as potential prognostic biomarkers of PDAC progression has not been investigated and could be important in stratifying patients for neoadjuvant and adjuvant therapy.

The main modality used for the staging of PDAC is cross-sectional imaging MDCT/MRI, preoperatively [23]. Staging of PDAC is based on the primary tumour itself, regional lymph node and distant metastases (TNM) staging system maintained by the American Joint Committee on Cancer (AJCC). Another common approach used to categorize PDAC is based on the resectability of the tumour, enabling the clinician to plan the most suitable treatment strategy, be it upfront surgery, neoadjuvant or palliative chemotherapy. Few studies have investigated prognostic biomarkers in the progression of PDAC and the role of circulating cytokines and angiogenic factors in advanced PDAC is worthy of investigation.

Cumulatively, the risk factors, presenting symptoms and circulating biomarkers associated with a diagnosis of PDAC may differ substantially from those that alter disease progression and survival. Identifying and understanding individual patient-related factors, risk factors, symptoms and potential biomarkers across the stages of PDAC could enable us to better target affected individuals to promote prompt health seeking behavior which, in turn, may result in an earlier diagnosis of PDAC at a resectable stage and improved survival of this devastating disease. No studies to date have identified and described the risk factors, symptoms and biomarkers of PDAC progression in patients from South Africa. Thus, in Black South African patients with a diagnosis of PDAC, the aim of our study was to assess the risk factor profile, presenting symptoms and potential prognostic biomarkers and determine their relationship, if any, with the stage of the cancer and/or metastatic PDAC.

## Materials and methods

### Ethical considerations

Ethical approval was obtained from the University of the Witwatersrand Human Research Ethics Committee (Medical) (clearance number M140669 and M160840) and from the

hospital's management of the Chris Hani Baragwanath Academic Hospital (CHBAH). The study was conducted according to the Helsinki declaration (2008 amended version) and written informed consent was obtained from each patient prior to enrollment.

## Patient selection and data collection

As part of a larger, ongoing prospective study in our Hepato-Pancreatico-Biliary (HPB) Unit at CHBAH, this study enrolled patients with a PDAC diagnosis, diagnosed either cytologically or histologically. Patients of self-reported African descent/heritage make up the vast majority of patients seen at this study site, and during the study period only patients of African descent were included. Fittingly, there is scarcity of PDAC biomarker studies in this population.

Clinical and demographic data were recorded on standardized questionnaires and included the patient's age at diagnosis, gender, risk factor data for self-reported smoking status, self-reported alcohol consumption, hypertension, diabetes mellitus, vascular disease, BMI and functional performances status. Current smokers and those that quit less than 15 years ago were classified as 'Smokers'; patients who had never smoked or those with more than 15 years of abstinence were classified as 'Non-smokers' [24]. Number of pack years were not reported. Alcohol consumption and usage was recorded by means of the reliable and validated CAGE questionnaire. Performance status scales are tools that attempt to quantify a patient's general well-being and activities of daily living. We used the Eastern Cooperative Oncology Group (ECOG) and the Karnosky scores, both of which facilitate the classification of a patient's functional impairment, effectiveness of therapies and the prognosis of the patient. Presenting symptoms and signs were recorded and included jaundice, loss of weight, abdominal pain, vomiting and ascites.

From our larger study cohort that investigated circulating biomarkers in 85 PDAC patients [22], a total of 72 patients had imaging for staging and were included in this study. We retrospectively retrieved the abdominal CT scans from these patients performed at initial presentation to determine the tumour location and TNM stage of the cancer and correlated these findings with the demographic, risk factor and symptom data recorded, as well as to the levels of circulating inflammatory cytokines and angiogenic factors measured. Proximal tumour lesions included head and uncinated lesions, whereas distal tumour lesions included neck, body and tail lesions.

## Sample collection and biomarker measurements

Standard laboratory blood tests, as well as blood tests routinely requested by PDAC-treating clinicians were recorded. Inflammatory cytokines and angiogenic factor measurements were conducted as previously described in our study [22, 25]. Briefly, plasma levels of 38 inflammatory cytokines and angiogenic factors were measured blindly and in duplicate using commercially available human magnetic multiplex screening assays according to the manufacturer's instructions (Bio-Plex Pro™ Human Cytokine 27-plex, Bio-Plex Pro™ Human Th17 3-Plex and customised R&D Systems Human magnetic luminex screening assays).

## Statistical analyses

Sstatistical analyses were conducted using STATA Version 14.2 suite of analytics software. Non-parametric Mann-Whitney U and Kruskal Wallis tests were used as appropriate to determine differences in clinical parameters among the stages of PDAC, as well as to discriminate metastatic patients (M1) to those without metastases (M0). Tumour marker and biomarker measurements are presented as median values and interquartile ranges (IQRs). Pearson's Chi-squared and Fishers' tests were used for analyses of categorical data and expressed as absolute

and relative frequencies. Bonferroni corrections were applied, where applicable, for multiple testing. A value of p≤0.05 was considered statistically significant.

Univariate and multivariate logistical regression analyses were conducted on logarithmic transformed values for risk analysis and model building, and the Hosmer-Lemeshow Test was applied to determine the goodness-of-fit of the multivariate models. Only variables with p-values < 0.2 in the univariate analyses were considered for multivariate models. Receiver Operating Characteristic (ROC) curve analysis was conducted, sensitivity and specificity data reported to determine optimal cut-off points for multivariate combined biomarkers in predicting metastasis. We report odds ratios with 95% confidence intervals to determine the effect size of the biomarkers in accurately discriminating metastasis in PDAC.

## Results

A total of 72 patients with confirmed PDAC were included in the study and the complete TNM staging of each tumour at diagnosis was determined. The mean (±SD) age at presentation was 60.3 (±11.1) years. Overall there was male predominance (56.9%), although females made up 56.4% of patients with metastatic disease. The majority of patients had stage 2 disease (36.1%), followed by stage 3 (30.6%), stage 4 (25.0%) and stage 1 (8.3%). Thus, over half of our PDAC patients (n = 40, 55.6%) had locally advanced and advanced disease stages 3 or 4. There were no significant differences in age between the four tumour stage groups (p = 0.440; Table 1).

**Table 1. Risk factors associated with PDAC and tumour location according to TNM stage and metastatic status (M0/M1).**

| Parameters | All PDAC (n = 72) | Stage 1 (n = 6) | Stage 2 (n = 26) | Stage 3 (n = 22) | Stage 4 (n = 18) | p-value | M0 vs M1 p-value |
|---|---|---|---|---|---|---|---|
| Age, years (mean ± SD) | 60.3 ± 11.1 | 64.1 ± 11.1 | 61.9 ± 10.2 | 57.1 ± 12.1 | 60.6 ± 11.2 | 0.44* | 0.94 |
| BMI, kg/m$^2$ (median, IQR) | 22.4 (18.6–25.3) | 20.9 (19.2–24.3) | 24.1 (21.0–27.8) | 18.8 (15.8–23.6) | 22.8 (19.5–24.8) | **0.05*** | 0.64 |
| Gender | | | | | | | |
| *Male*, n (%) | 41 (56.9%) | 5 (83.3%) | 15 (57.7%) | 13 (59.1%) | 8 (44.4%) | 0.43 | 0.28 |
| | **All PDAC** *(n = 66)* | **Stage 1** *(n = 6)* | **Stage 2** *(n = 23)* | **Stage 3** *(n = 19)* | **Stage 4** *(n = 18)* | **p-value** | **M0 vs M1 p-value** |
| Smoking status, n (%) | | | | | | | |
| *Smokers* | 34 (51.5%) | 5 (83.3%) | 10 (43.5%) | 11 (57.9%) | 8 (44.4%) | | |
| *Non-smokers* | 32 (48.5%) | 1 (16.7%) | 13 (56.5%) | 8 (42.1%) | 10 (55.6%) | 0.32** | 0.58 |
| Alcohol, n (%) | | | | | | | |
| *Yes* | 46 (70.8%) | 5 (83.8%) | 14 (63.6%) | 14 (73.7%) | 13 (72.2%) | 0.78 | 0.87 |
| CAGE score, n (%) | *(n = 55)* | *(n = 5)* | *(n = 20)* | *(n = 14)* | *(n = 16)* | | |
| 0 –not dependent | 36 (65.5%) | 2 (40.0%) | 12 (60.0%) | 10 (71.4%) | 12 (75.0%) | 0.31 | 0.32 |
| 1 | 6 (10.9%) | 1 (20.0%) | 0 | 2 (%) | 3 (18.8) | | |
| 2 | 2 (3.6%) | 0 | 2 (10.0%) | 0 | 0 | | |
| 3 | 8 (14.6%) | 2 (40.0%) | 4 (20.0%) | 1 (7.1%) | 1 (6.3) | | |
| 4 | 3 (5.5%) | 0 | 2 (10%) | 1 (7.1%) | 0 | | |
| Hypertension, n (%) | 26 (39.4%) | 2 (33.3%) | 11 (47.8%) | 6 (31.6%) | 7 (38.9%) | 0.77 | 0.60 |
| Diabetes Mellitus, n (%) | 16 (24.6%) | 1 (16.7%) | 5 (22.7%) | 6 (31.6%) | 4 (22.2%) | 0.87 | 0.55 |
| Vascular Disease, n (%) | 2 (3.1%) | 0 | 2 (9.1%) | 0 | 0 | 0.44 | 0.53 |
| Tumour location, n (%) | *(n = 69)* | *(n = 6)* | *(n = 24)* | *(n = 22)* | *(n = 17)* | | |
| *Proximal lesions* | 63 (91.3%) | 6 (100%) | 23 (95.8%) | 18 (81.8%) | 16 (94.1%) | 0.38 | >0.99 |
| *Distal lesions* | 6 (8.7%) | 0 | 1 (4.2%) | 4 (18.2%) | 1 (5.9%) | | |

*Abbreviations*: BMI, body mass index; M0, no metastasis; M1 metastatic PDAC; SD, standard deviation.

*Kruskal-Wallis ANOVA.

**Chi$^2$/Fisher's exact for overall smoking status.

Table 1 also shows the risk factors present within each TNM-stage group of patients. Borderline significant differences between the BMI measurements across the TNM-stages is shown (p = 0.05). Notably, only 12.5% of our PDAC patients were obese with BMI $\geq$ 30 kg/m$^2$.

Just over half of the patients were smokers (51.5%) and, overall, smoking status did not show any association with PDAC stage (p = 0.32) or metastatic status (p = 0.58). There was a significant difference between smoking status and gender (p<0.0001); specifically, only 21.6% of males were non-smokers versus 82.8% of females. Moreover, 90% of females with metastatic PDAC were non-smokers compared to 12.5% of males with metastatic disease (p = 0.003). There was no significant difference in age at presentation based on smoking status (data not shown).

Interestingly, where the majority of PDAC patients consumed alcohol (70.8%), just over a third (34.5%) were alcohol dependent as per the CAGE score analysis (Table 1). For stage 1 disease, 60% of PDAC patients had some level of alcohol dependence compared to only 25% for stage 4 disease. Overall, the prevalence of hypertension, diabetes mellitus and vascular disease in this PDAC cohort was 39.4%, 24.6% and 3.1%, respectively (Table 1), and showed no significant differences across PDAC stages.

The tumour location in the majority of patients was proximal lesions (91.3%) and only 8.7% were distal lesions. There were no statistical significant differences in the location of the lesion across the stages of PDAC or according to metastatic status (Table 1). The majority of distal lesions (83.3%) were locally advanced and advanced disease stages 3 or 4, compared to 53.1% of in patients with proximal lesions, although this did not reach statistical significance probably due to small numbers and hence a Type II error.

The presenting symptoms/signs in PDAC patients according to stage are shown in Table 2. The most prevalent associated symptom was abdominal pain and present in 66.7% of all PDAC patients with higher percentages reported for more advanced stage patients. Abdominal pain in early disease is further discussed below, along with CRP and total bilirubin levels. The next commonest symptom was the presence of jaundice (64.6%) which was more common in early stage cancers, the latter of which were all proximal tumour lesions. Of note, 56.5% of PDAC patients from our larger study cohort in our diagnostic biomarker study presented with obstructive jaundice [22]. Loss of weight, vomiting and ascites were present in 50.0%, 21.2%

**Table 2. Associated symptoms/signs in PDAC according to TNM stage and metastatic status.**

| Parameters present | All PDAC | Stage 1 | Stage 2 | Stage 3 | Stage 4 | p-value* | M0 vs M1 p-value* |
|---|---|---|---|---|---|---|---|
| Jaundice, n (%) | 42 (64.6%) | 5 (83.3%) | 15 (68.2%) | 10 (52.6%) | 12 (66.7%) | 0.57 | 0.54 |
| *Proximal lesions* | *95.1%* | *100%* | *100%* | *90%* | *90.9%* | | |
| *Distal lesions* | *4.9%* | *0%* | *0%* | *10%* | *9.1%* | | |
| Loss of weight, n (%) | 33 (50.0%) | 3 (50.0%) | 11 (47.8%) | 9 (47.4%) | 10 (55.6%) | 0.97 | 0.39 |
| *Proximal lesions* | *93.0%* | *100%* | *100%* | *77.8%* | *100%* | | |
| *Distal lesions* | *7.0%* | *0%* | *0%* | *22.2%* | *0%* | | |
| Abdominal pain, n (%) | 44 (66.7%) | 3 (50.0%) | 14 (60.9%) | 14 (73.7%) | 13 (72.2%) | 0.61 | 0.39 |
| *Proximal lesions* | *93.0%* | *100%* | *100%* | *78.6%* | *100%* | | |
| *Distal lesions* | *7.0%* | *0%* | *0%* | *21.4%* | *0%* | | |
| Vomiting, n (%) | 14 (21.2%) | 1 (16.7%) | 5 (21.7%) | 3 (15.8%) | 5 (27.8%) | 0.85 | 0.32 |
| Ascites, n (%) | 4 (6.1%) | 0 | 0 | 2 (10.5%) | 2 (11.1%) | 0.34 | 0.30 |

*Abbreviations*: M0, no metastasis; M1 metastatic PDAC.

*Fisher's exact test.

**Table 3. ECOG performance in PDAC patients according to TNM stage and metastatic status.**

| ECOG scale | All PDAC | Stage 1 | Stage 2 | Stage 3 | Stage 4 | p-value* | M0 vs M1 p-value |
|---|---|---|---|---|---|---|---|
| 0 –asymptomatic, fully functional | 3 (4.8%) | 0 | 2 (9.1%) | 0 | 1 (5.6%) | 0.48 | 0.25 |
| 1 –symptomatic but ambulatory, able to do light work | 43 (68.3%) | 3 (50.0%) | 13 (59.1%) | 13 (76.5%) | 14 (77.8%) | | |
| 2 –capable of selfcare, unable to work, <50% of day in bed | 15 (23.8%) | 3 (50.0%) | 6 (27.3%) | 3 (17.7%) | 3 (16.7%) | | |
| 3– limited selfcare only, >50% of day in bed | 2 (3.2%) | 0 | 1 (4.6%) | 1 (5.9%) | 0 | | |

*Abbreviations*: ECOG, Eastern Cooperative Oncology Group; M0, no metastasis; M1 metastatic PDAC.

*Kruskal-Wallis ANOVA.

and 6.1% of patients, respectively. Ascites was only present in patients with advanced stage 3 and 4 cancers.

Both the Karnofsky and ECOG performance scales were used to determine the functional status of our PDAC study patients. In our institution we routinely use the ECOG performance scale and we therefore subsequently analyzed functional status according to PDAC stage using the ECOG scale alone (Table 3). There were no significant differences between the ECOG scores and the stage of the disease (p = 0.48). The majority of PDAC patients in our study (68.3%) had an ECOG score of 1, i.e. they were symptomatic, yet ambulatory and able to do light work. This held true at each stage of the cancer. Interestingly, only two patients had the highest ECOG score of 3 (capable of limited self-care and confined to the bed for more than 50% of waking hours) and they were at stage 2 and 3. The latter, demonstrates that those in the more advanced stages do not necessary present with a higher ECOG score, as would be expected. Notably, both patients with ECOG scores of 3 had raised CRP levels.

Table 4 shows the biomarker measurements in PDAC patients according to TMN stage.

A significantly lower platelet count, albeit within the normal range, is associated with advanced PDAC. Specifically, platelet counts were significantly lower in stage 4 PDAC patients when compared to those in stage 1 (p = 0.01), stage 2 (p = 0.01) or stage 3 (p = 0.04) (Mann-Whitney U tests). Furthermore, the significance becomes stronger when metastatic patients are compared to those without metastasis (p = 0.005).

CRP levels were significantly higher in stage 4 PDAC patients compared to stage 2 (106.0 vs 36.5 mg/L, respectively; p = 0.01, Mann-Whitney U test) and in metastatic patients compared to those without metastasis (106.0 vs 49.5 mg/L, respectively; p = 0.037). Moreover, none of the metastatic patients had CRP levels <27 mg/L, compared to 54% in non-metastatic patients (data not shown).

Even though there was no significant difference for total bilirubin levels across the stages, median values for stage 1 disease were significantly higher when compared to stage 3 disease specifically (p = 0.02, Mann-Whitney U test). This is in keeping with the higher prevalence of jaundice in stage 1 disease with proximal lesions only compared to stage 3 disease at 83.3% and 52.6%, respectively (Table 2). Although a diagnosis of cholangitis was not recorded in this study cohort, the average CRP and total bilirubin levels in the stage 1 patients of >77 mg/L and >324 μmol/L, respectively, may be indicative of cholangitis. The latter may also explain the higher prevalence of abdominal pain in these early stage cancers.

Circulating GGT, is borderline significantly raised with advanced PDAC in this study population (p = 0.06), but failed to discriminate metastatic PDAC from non-metastatic disease. HbA1C levels were within normal limits throughout all the stages even though diabetes mellitus is present in a quarter of our study patients. However, we did not take treatment in this latter group of patients into account. The tumour marker CA19-9 was significantly elevated in

**Table 4. Biomarker measurements in PDAC patients according to TNM stage and metastatic status.**

| Parameters | All PDAC | Stage 1 | Stage 2 | Stage 3 | Stage 4/M1 | p-value* | M0 vs M1 p-value |
|---|---|---|---|---|---|---|---|
| Platelet count ($10^9$/L) | 315.0 (249.0–402.0) | 357.5 (324.0–507.0) | 325.0 (273.0–398.0) | 314.5 (250.0–470.0) | 248.5 (208.0–316.0) | **0.025** | **0.005** |
| CRP (mg/L) | 52.5 (18.5–116.0) | 101.5 (69.0–134.0) | 36.5 (12.0–86.0) | 49.0 (13.0–116.0) | 106.0 (30.0–150.0) | 0.069 | **0.037** |
| Total bilirubin (μmol/L) | 136.0 (29.0–312.0) | 301.0 (212.0–426.0) | 126.0 (33.0–380.0) | 96.5 (10.0–269.0) | 218.5 (36.0–307.0) | 0.110 | 0.619 |
| GGT (units/L) | 443.0 (143.0–777.0) | 473.0 (441.0–1183.0) | 255.0 (95.0–502.0) | 471.0 (136.0–845.5) | 555.0 (383.0–942.0) | 0.057 | 0.069 |
| HbA1c (%) | 5.9 (4.9–7.3) | 4.9 (4.6–9.3) | 6.3 (4.8–8.5) | 6.2 (4.6–7.7) | 5.7 (5.3–5.9) | 0.846 | 0.517 |
| CA19-9 (U/mL) | 403.0 (37.5–5660.0) | 1614.5 (34.0–214185) | 97.2 (32.0–198) | 1303.8 (9.9–13102) | 5660 (403–13408) | **0.039** | **0.018** |
| CEA (ng/mL) | 6.2 (3.1–12.9) | 4.6 (3.3–5.9) | 6.0 (3.0–9.1) | 16.8 (7.2–41.3) | 4.3 (2.9–10.3) | 0.150 | 0.281 |
| *Potential biomarkers*: | | | | | | | |
| IL-15 (pg/mL) | 12.0 (0.73–32.7) | 0.73 (0.73–32.7) | 9.99 (0.73–37.3) | 30.7 (13.7–45.4) | 0.73 (0.73–18.8) | **0.036** | **0.041** |
| GM-CSF (pg/mL) | 35.0 (20.8–58.0) | 59.9 (28.3–65.6) | 34.4 (18.9–62.1) | 44.1 (28.5–53.5) | 26.5 (12.2–45.8) | 0.111 | **0.024** |
| VEGF-R2 (pg/mL) | 8088 (6077–11177) | 7272 (5544–8294) | 9302 (6900–12147) | 5481 (3083–7969) | 9551 (7103–11145) | **0.048** | 0.123 |

Values are presented as median (interquartile range [IQR])

*Kruskal-Wallis ANOVA. *Abbreviations*: CA19-9, carbohydrate or cancer antigen 19–9; CEA, carcinoembryonic antigen; CRP, **C-reactive protein**; GGT, gamma-glutamyl transferase; HbA1c, glycated hemoglobin A1c.

advanced PDAC disease with the highest levels in those with stage 4 disease (p = 0.039) and significant differences between metastatic vs non-metastatic PDAC (p = 0.018).

Of the 38 circulating angiogenic factors and inflammatory cytokines measured as potential prognostic biomarkers, only IL-15, GM-CSF and VEGF-R2/KDR showed significantly altered levels with advanced disease (Table 4).

Univariate logistic regression analysis further supported CRP, GGT, CA19–9 and VEGF-R2 as independent predictors of metastatic PDAC, whilst platelet counts, IL-15 and GM-CSF levels were independent negative predictors of metastasis in our study cohort (Table 5). Moreover, multivariate logistic regression analysis showed improved areas under the ROC curve (AUCs) when CA19-9 was combined with negatively associated biomarkers in predicting PDAC metastasis (Fig 1). A combined panel of CA19-9, IL-15 and platelet count increased the AUC from 0.74 for CA19-9 alone and correctly classifying 69.8% of patients, to 0.87 and correctly classifying 76.5% of patients.

CA19-9 aside, when using specific cut-off values for our independent negative predictors against metastatic PDAC in univariate and multivariate logistic regression analyses, the effect size in predicting metastatic PDAC is statistically highly significant (Table 6). Specifically, using cut-off values for platelet counts of $\leq 290 \times 10^9$/L, IL-15 $\leq 2.4$ pg/mL and GM-CSF $\leq 50$ pg/mL as

**Table 5. Univariate logistic regression analyses to predict metastasis in PDAC.**

| Effect | Odds Ratio* | OR 95% CI | P-value |
|---|---|---|---|
| Platelet count ($10^9$/L) | 0.021 | 0.001–0.491 | **0.006** |
| CRP (mg/L) | 3.74 | 1.04–13.5 | **0.023** |
| GGT (units/L) | 3.43 | 0.93–12.6 | **0.045** |
| CA19-9 (U/mL) | 1.72 | 0.99–2.99 | **0.041** |
| IL-15 (pg/mL) | 0.439 | 0.207–0.929 | **0.024** |
| GM-CSF (pg/mL) | 0.146 | 0.026–0.817 | **0.009** |
| VEGF-R2 (pg/mL) | 21.0 | 0.68–649 | **0.045** |

*Modelled probability that the patient has metastasis. Statistics conducted on log-transformed values.

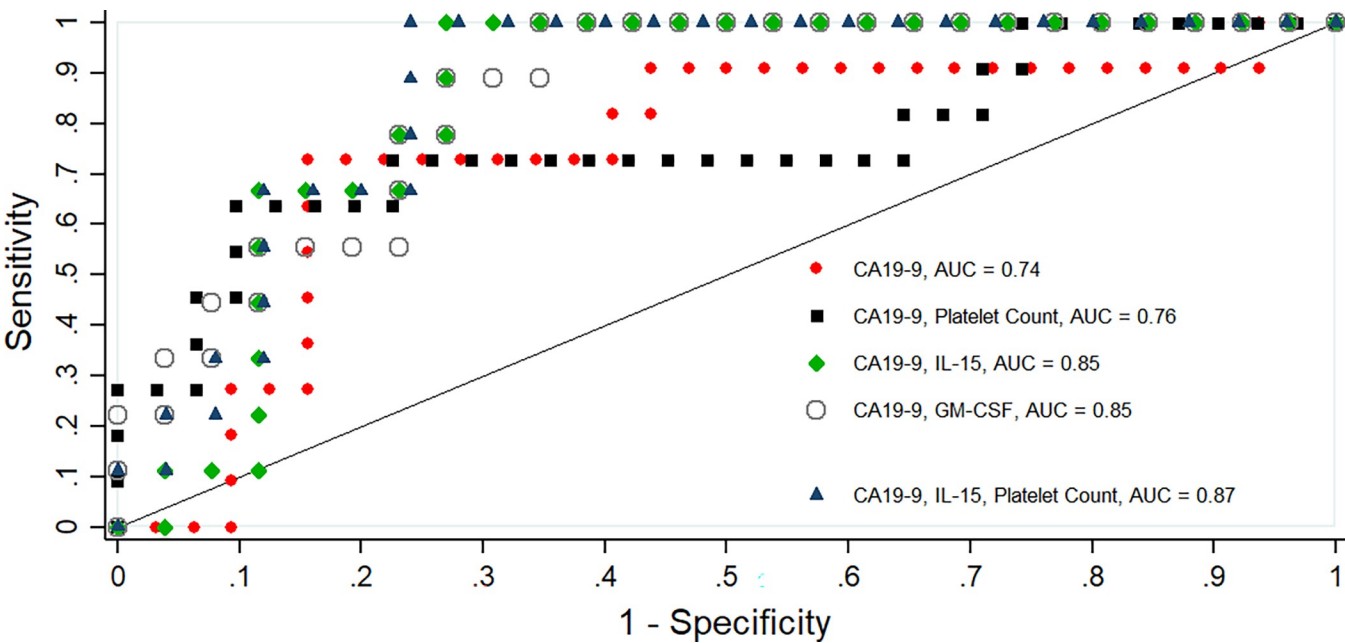

**Fig 1. Comparison of ROC curves for combined biomarkers vs CA19-9 in determining PDAC metastasis.**

determined by sensitivity and specificity analyses, the odds ratio of the combined biomarkers indicate that a patient is on average 21 times more likely to have metastatic PDAC, ranging from 214% to 221 times more likely (p < 0.0001). The high specificity indicates that there is a 95.4% probability that you will have all three values above these cut-off levels if you do not have metastatic PDAC (true negative rate), but notably this is at a low sensitivity of 50%.

## Discussion

PDAC is an aggressive cancer and the high risk of metastatic disease, in addition to the late presentation, contribute to the poor prognosis in these patients. We describe potential prognostic indicators of advanced disease in this first study from South Africa on the risk factors, symptoms and circulating biomarkers according to the TNM stage and metastatic status in PDAC patients.

**Table 6. Univariate and multivariate logistic regression analyses to predict metastasis in PDAC with specific cut-off values.**

| Effect | Odds Ratio* | OR 95% CI | P-value | Sensitivity | Specificity | PPV | NPV | Correctly classified |
|---|---|---|---|---|---|---|---|---|
| *Univariates with cut-off values:* | | | | | | | | |
| Platelets $\leq$ 290 10$^9$/L | 6.01 | 1.62–24.6 | **0.002** | 72.2% | 69.8% | 44.4% | 88.3% | 70.4% |
| IL-15 $\leq$ 2.4 pg/mL | 4.40 | 1.29–15.0 | **0.014** | 68.8% | 66.7% | 40.7% | 86.5% | 67.2% |
| GM-CSF $\leq$ 50.0 pg/mL | 12.0 | 1.46–98.8 | **0.002** | 93.8% | 44.4% | 36.0% | 95.5% | 56.8% |
| *Multivariates with cut-off values:* | | | | | | | | |
| Platelets $\leq$290, IL-15 $\leq$ 2.4 | 10.0 | 1.99–54.5 | **<0.001** | 50.0% | 90.9% | 64.7% | 84.5% | 80.7% |
| Platelets $\leq$290, GM-CSF $\leq$ 50 | 13.0 | 2.74–64.7 | **0.0001** | 62.5% | 88.6% | 64.7% | 87.6% | 82.1% |
| **Platelets $\leq$290, IL-15 $\leq$ 2.4, GM-CSF $\leq$ 50** | 21.0 | 3.14–221.0 | **<0.0001** | 50.0% | 95.4% | 78.6% | 85.1% | 84.1% |

*Modelled probability that the patient has metastasis. Statistics conducted on log-transformed values.

## Presenting symptoms and risk factors

In our study, the mean age at presentation of patients diagnosed with PDAC was 60.3 years, which is slightly younger than that reported from national data for PDAC [6], and in contrast to the United States who report a median age at diagnosis of 71 years [26]. We showed no remarkable differences in mean age across the stages of PDAC.

We report an overall male dominance of 56.9% in this study population, however female predominance of 56.4% was found in metastatic PDAC. Interestingly, female predominance in metastatic PDAC reportedly depends on the site of metastases: specifically, major differences in patient profiles were reported for "lung-only" vs "liver-only" metastases, with Claire's group in 2016 and Arnaoutakis' group in 2011 reporting female predominance for "lung-only" metastatic PDAC of 73% and 55%, respectively [27, 28]. The site of metastases has shown prognostic value in breast cancer survival [29] and should be included in future prognostic PDAC studies.

From the literature, smoking in itself is a major risk factor for developing PDAC and the pattern of smoking duration or time since quitting, is more relevant than the smoking intensity [26, 30]. In our study, just over half of our patients (51.5%) were smokers, and 44.4% of metastatic patients were smokers. Notably, the vast majority of the smokers in our study were male (84.9%). Regardless, the majority of metastatic PDAC patients were females of whom 90% were non-smokers, compared to 12.5% of males with metastatic disease. The latter may reflect that, even though smoking is a risk factor for developing PDAC, if may not necessarily increase metastatic risk in males, yet additional, unidentified risk factors may be at play in female patients in our local population, and warrants further investigation.

Alcohol consumption is notably higher in our study population (70.8%) than the 31% reported for South Africans aged 15 years and older by the WHO in 2016. From the CAGE questionnaire two thirds of our PDAC patients did not demonstrate alcohol dependence. The level of alcohol consumption was not quantified in our study and therefore high daily consumptions of >40 grams per day could not be assessed. No significant differences were seen for alcohol consumption and PDAC stage.

Obesity, as defined by BMI of 30 kg/m$^2$ or above, is a known risk factor for PDAC and 12.5% of our study patients were obese. Nonetheless, the median BMI was normal at each TNM stage and, as we only had BMI levels at presentation, we are unable to report on the effect of PDAC on BMI over the course of onset to presentation, or whether obesity is a risk factor for PDAC in our population.

The association between PDAC and diabetes mellitus has been widely reported in the literature and the prevalence of PDAC is significantly higher in adults with new-onset diabetes mellitus than that in the general population [31, 32]. In our study, 24.6% of patients with PDAC had diabetes, which is almost double that of the 12.8% prevalence reported for South Africa by the International Diabetes Federation and the 14.3% prevalence reported for this local Johannesburg population [33]. Unfortunately, our study is limited in that we are unable to report on the duration of diabetes, and thus on new-onset diabetes mellitus in these patients.

The prevalence of hypertension in South Africans aged between 40 and 60 years has been reported as >40% [34]. Specifically, the prevalence in the local Johannesburg population of Soweto is estimated to be 54.1%. In our study, hypertension was present in 39.4% of PDAC patients and this did not differ according to PDAC stage. Similarly, vascular disease was not a major presenting risk factor, nor was it associated with stage or metastatic disease.

The most common presenting symptoms recorded in our study population were abdominal pain, jaundice and loss of weight. Where the prevalence of abdominal pain and jaundice were in line with reported literature [6], loss of weight in half of our study population was lower

than reported elsewhere [14], and not associated with having diabetes in our population. Overall, all of the presenting symptoms investigated were unrelated to the stage of PDAC in our study.

## Circulating biomarkers as prognostic indicators of advanced PDAC

Platelet counts, IL-15 and GM-CSF were the most significant independent predictors of metastatic PDAC in our study. Notably, the platelet count is significantly and inversely associated with advancing stages of PDAC and became more pronounced in metastatic PDAC vs non-metastatic disease patients. In addition, this association remained significant in each of the individual T-, N- and M-staging categories (data not shown). The contribution of platelet counts and thrombocytosis in pancreatic cancer prognosis is still controversial [21, 35]. As established in lung, breast and gastrointestinal cancers [36–40], some studies have shown that increased pre-operative platelet counts or thrombocytosis have a poorer prognosis in PDAC [18, 35, 41], Conversely, others have shown that pre-operative platelet counts may have different prognostic implications in PDAC than in other cancers, with improved prognosis in patients with increased platelet counts [17, 20]. One hypothesis has been that PDAC patients may present with lower platelet counts as a result of occlusion of the splenic vein from tumour invasion leading to a splenomegaly mediated thrombocytopenia, and another that extremely aggressive PDAC might metastasize to the bone marrow leading to thrombocytopenia [35]. From our study, reduced platelet counts, albeit within normal ranges, is an independent predictor of metastatic PDAC and the underlying mechanism is worthy of investigation. Where the prognostic value of a pre-treatment platelet count may depend on the site of metastasis, and the latter should be reported in future prognostic studies, future studies including biochemical mechanisms of platelet activation and the role of platelets in shifting the circulating tumor cell phenotype from an epithelial to a mesenchymal-like cell, as well as the state and predominance of these platelet-tumor cell aggregates, are required to elucidate the prognostic value of platelets in PDAC [42]. Certainly the translation of the latter investigations into routine laboratory assays have not yet been fulfilled [43].

The significance of the pro-inflammatory cytokine IL-15 in PDAC is in line with our previous diagnostic biomarker work [22]; IL-15 is significantly and inversely associated with advancing stages of PDAC, in addition to being a significant independent predictor of metastatic PDAC. IL-15's anti-tumour response is well documented and The American National Cancer Institute identified it as one of the most promising cancer immunotherapy targets [44].

The role of GM-CSF in clinical oncology is still highly controversial. We report a significant, inverse relationship between GM-CSF and advanced PDAC, thereby supporting the role of GM-CSF as an immune stimulant that enhances the antitumor immune response. In contrast, GM-CSF has also been described as an immune-independent, tumor-promoting factor by promoting the growth and migration of tumor cells in multiple cancer types. Nevertheless, the underlying mechanisms of these controversial interactions between tumour cells and GM-CSF has not been elucidated and remains an interesting area of cancer research [45].

By means of multivariate logistic regression analysis, we built a combined biomarker model of these three circulating biomarkers (platelet count, IL-15 and GM-CSF) that discriminates metastatic PDAC from non-metastatic disease. From sensitivity/specificity data, we identified and used specific cut-off values for platelet counts of $\leq290\text{x}10^9$/L, IL-15 $\leq2.4$ pg/mL and GM-CSF $\leq50$ pg/mL. Using this combined biomarker panel, the odds ratio indicates that a PDAC patient is on average 21 times more likely to have metastatic PDAC if their biomarker levels fall below these three marker cut-offs, which is at least 214% more likely or up to 221 times more likely (p < 0.0001). The high specificity indicates that there is a 95.4% probability

that a PDAC patient will test negative for all those cut-offs values (i.e. have all values above) if they do not have metastatic PDAC (true negative rate), and hence the likelihood of a positive test being correct is very high but notably this is at a low sensitivity of 50%. Unfortunately, the latter renders this combined biomarker as not very useful clinically, as 50% of metastatic PDAC could be missed. However, it gives valuable information and raises important questions with regard to metastasis biology, and could potentially be useful in determining subclinical risk.

CRP, CA19-9 and GGT were also independent predictors of metastatic PDAC in our study, albeit to a lesser extent that the aforementioned circulating biomarkers. Raised CRP levels in different cancer patients have been associated with poorer survival outcomes, lower functional activity, abnormal metabolism and more extensive disease, and specifically in PDAC, elevated CPR levels reflect a more aggressive tumour [46]. In our study, CRP was an independent predictor of metastatic PDAC. Moreover, none of the metastatic patients had CRP levels <27 mg/L, compared to 54% in non-metastatic patients (data not shown). Similarly, Aktekin's group (2019) reported significantly raised CRP levels in irresectable vs reseactable PDAC, as well as in patients with radiological lymph node metastasis compared to node negative disease [47]. However, the mechanism behind this requires investigation. Notably, both patients in our study with limited functional performance had high CRP levels, in line with the literature [46].

Again, CA19-9 and GGT are independent predictors of metastasic PDAC, and whether these markers are involved in platelet-tumour cell interaction has not been investigated. The high CA19-9 in stage 1 patients is out of keeping with what is expected and may be due to either the small number of patients in this stage group, or the impact of obstructive jaundice on CA19-9. It would be worthwhile investigating the prognostic value of CA19-9 once the jaundice has been relieved, although this was not within the scope of this study. Where accumulating evidence suggests a strong association between elevated GGT levels and cancer occurrence and progression, the role of GGT as an indicator of PDAC risk is weak [48]. A recent study suggests that GGT could be a potential biomarker of survival in metastatic PDAC, even though the underlying mechanism is unknown [49].

Our study is not without limitations. Due to all parameters in this study measured at presentation, we are unable to analyse the longitudinal effect that many factors have on PDAC, such as hypertension and diabetes, or the effect that PDAC has on many variables, such as BMI. Furthermore, we were unable to quantify many categorical variables recorded, in terms of quantifying amount of alcohol ingested daily, pack years of smoking, the onset and new onset of diabetes mellitus, patients BMI prior to a PDAC diagnosis. Significant results from some of the subgroup analyses may be missed due to small numbers, particularly for some categories analyzed according to stage in Tables 2 and 3. Hence, we need to further validate this study in an independent validation cohort prior to employing the information in clinical decision making.

In conclusion, our study is the first of its kind to identify and describe the risk factors, symptoms and potential prognostic biomarkers of PDAC progression and metastatic PDAC in patients from South Africa. Our PDAC population is younger than those in other national and international studies and we report female predominance in metastatic PDAC. Where presenting symptoms and risk factors are not good predictors of metastatic PDAC in our population, we have identified circulating biomarkers that are significant prognostic indicators of metastatic PDAC, specifically decreased levels of platelets, IL-15 and GM-CSF, an increased levels of CA19-9, CRP and GGT. A combined biomarker panel of platelet count, IL-15 and GM-CSF is a significant indicator of metastatic PDAC, showing that a patient is on average 21 times more likely to have metastatic PDAC if their biomarker levels fall below the marker cut-off levels. However, the clinical utility of these biomarkers, such as in stratifying patients for neoadjuvant therapy, will have to be addressed in future prospective studies.

## Acknowledgments

The authors express their sincere thanks Dr Zafar Khan for his assistance with data collection.

## Author Contributions

**Conceptualization:** Deirdré Kruger, Martin Smith.

**Data curation:** Deirdré Kruger, Nicola Lahoud, John Devar.

**Formal analysis:** Deirdré Kruger, Nicola Lahoud, John Devar.

**Funding acquisition:** John Devar, Martin Smith.

**Investigation:** Deirdré Kruger, Nicola Lahoud, Yandiswa Y. Yako, Martin Smith.

**Methodology:** Deirdré Kruger, Nicola Lahoud, Yandiswa Y. Yako, John Devar.

**Project administration:** Deirdré Kruger.

**Resources:** Deirdré Kruger.

**Supervision:** Deirdré Kruger, John Devar, Martin Smith.

**Validation:** Deirdré Kruger, John Devar.

**Visualization:** Deirdré Kruger.

**Writing – original draft:** Deirdré Kruger, Nicola Lahoud.

**Writing – review & editing:** Deirdré Kruger, Nicola Lahoud, Yandiswa Y. Yako, John Devar, Martin Smith.

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
