## [Decision Letter · Decision Letter 0]

24 Dec 2021

PANCREATIC DUCTAL ADENOCARCINOMA: PROGNOSTIC INDICATORS OF ADVANCED DISEASE

PONE-D-21-20556

Dear Dr. Kruger,

We’re pleased to inform you that your manuscript has been judged scientifically suitable for publication and will be formally accepted for publication once it meets all outstanding technical requirements.

Kind regards,

Hans A. Kestler

Academic Editor

PLOS ONE

Additional Editor Comments (optional):

Reviewers' comments:

Reviewer's Responses to Questions

**Comments to the Author**

1. Is the manuscript technically sound, and do the data support the conclusions?

Reviewer #1: Yes

Reviewer #2: Yes

2. Has the statistical analysis been performed appropriately and rigorously? 

Reviewer #1: I Don't Know

Reviewer #2: Yes

3. Have the authors made all data underlying the findings in their manuscript fully available?

Reviewer #1: Yes

Reviewer #2: Yes

4. Is the manuscript presented in an intelligible fashion and written in standard English?

Reviewer #1: Yes

Reviewer #2: Yes

5. Review Comments to the Author

Reviewer #1: The article addresses an important issue that is of high clinical relevance. The number of subjects in each group reflects the clinical reality, which causes the unequal number of groups. The markers studied are mostly easy to determine and support a high clinical impact.

Reviewer #2: The study aims at assessing the risk factor profile, presenting symptoms and circulating biomarkers for pancreatic ductal adenocarcinoma (PDAC) according to the TNM stage and metastatic status in a cohort of 72 PDAC patients. This is the first study in a South African population and apart of clinical and demographic data, the authors subjected blood specimens to the analysis of 38 inflammatory cytokines and angiogenic factors. Within others, they report an overall male (i.e. with PDAC) dominance, however with a female predominance in metastatic PDAC (depending on the site of metastasis). The authors reveal that while smoking represents a risk factor for developing PDAC, it may not necessarily increase the metastatic risk in males. In this study, decreased levels of platelet counts, IL-15 and GM-CSF proved as the most significant independent predictors of metastatic PDAC. To discriminate metastatic PDAC from non-metastatic disease, the authors built a combined biomarker (i.e. platelet counts, IL-15 and GM-CSF) model. Importantly, this biomarker panel shows that a patient is prone on average 21x higher to develop a metastatic PDAC if the biomarkers fall below the cut-off levels. While this panel may not render immediate clinical use (given the around 50% missed metastatic PDACs), the authors emphasize its importance in delineation of subclinical risk. Although to a lesser extent in this study, increased levels of CRP, CA19-9 and GGT also proved as metastatic PDAC predictors. The authors also point to the limitations of the study (e.g. lack of analysis of hypertension or diabetes as factors that may impact on PDAC) or the lack of record on categorical variables (e.g. amount of alcohol ingested daily).

Altogether, this is a carefully conducted study and well written & presented manuscript. No concerns could be raised. Therefore, I recommend the publication in PlosOne.

6. PLOS authors have the option to publish the peer review history of their article (what does this mean?). If published, this will include your full peer review and any attached files.

Reviewer #1: No

Reviewer #2: **Yes: **Alexander Kleger

---

## [Editor Report · Acceptance letter]

31 Dec 2021

PONE-D-21-20556 

PANCREATIC DUCTAL ADENOCARCINOMA: PROGNOSTIC INDICATORS OF ADVANCED DISEASE 

Dear Dr. Kruger:

I'm pleased to inform you that your manuscript has been deemed suitable for publication in PLOS ONE. Congratulations! Your manuscript is now with our production department. 

Kind regards, 

on behalf of

Prof. Hans A. Kestler 

Academic Editor

PLOS ONE